# Enhancement of Plumbagin Production through Elicitation in In Vitro-Regenerated Shoots of *Plumbago indica* L.

**DOI:** 10.3390/plants13111450

**Published:** 2024-05-23

**Authors:** Yaowapha Jirakiattikul, Srisopa Ruangnoo, Kanokwan Sangmukdee, Kornkanok Chamchusri, Panumart Rithichai

**Affiliations:** 1Department of Agricultural Technology, Faculty of Science and Technology, Thammasat University, Pathum Thani 12120, Thailand; kanokwan.sang@dome.tu.ac.th (K.S.); kornkanok.chamc@dome.tu.ac.th (K.C.); panumart@tu.ac.th (P.R.); 2Department of Applied Thai Traditional Medicine, Faculty of Medicine, Thammasat University, Pathum Thani 12120, Thailand; srisopar@tu.ac.th

**Keywords:** elicitor, *Plumbago indica*, regenerated shoot, secondary metabolites, yeast extract

## Abstract

*Plumbago indica* L. contains a valuable bioactive compound called plumbagin. Elicited regenerated shoots grown in vitro could be another source of high-yielding plumbagin. The purpose of this investigation was to examine the effects of elicitor type and concentration, as well as elicitation period, on plumbagin content in in vitro-regenerated shoots of *P. indica*. Nodal explants were cultured on Murashige and Skoog (MS) medium containing 1 mg/L benzyladenine (BA) in combination with 0–150 mg/L yeast extract or 50–150 µM salicylic acid for four weeks. Plumbagin levels of 3.88 ± 0.38% and 3.81 ± 0.37% *w*/*w* g dry extract were achieved from the 50 and 100 mg/L yeast extract-elicited shoots, which were higher than the value obtained for the control. However, the addition of salicylic acid did not increase the plumbagin content. In the elicitation period experiment, nodal explants were cultured on MS medium supplemented with 1 mg/L BA and 50 mg/L yeast extract for durations of three, four and five weeks. The 4-week yeast extract-elicited shoot had a maximum plumbagin content of 3.22 ± 0.12% *w*/*w* g dry extract, greater than that of the control. In summary, the plumbagin content of the in vitro *P. indica* shoots was enhanced by 4-week elicitation using 50 mg/L yeast extract.

## 1. Introduction

*Plumbago indica* L. or *P. rosea* L. is classified within the family of Plumbaginaceae. It is naturally found in regions with warm to tropical climates across the world [1]. This medicinal plant species is a perennial herb with an upright stem reaching a height of 1–1.5 m, and an inflorescence that is 10–30 cm long with red-colored florets [1,2]. Fresh roots exhibit a light brown color, but when they dry out, they become dark or black [3]. The roots of this plant species have been used for treating cancer, dysmenorrhea, rheumatism, paralysis, leukoderma and rheumatoid arthritis [1] and as an abortifacient [2]. Thai traditional medicine doctors have utilized *P. indica* roots with another four medicinal plants, *Piper chaba* fruits, *P. sarmentosum* roots, *P. interruptum* stems and *Zingiber officinale* rhizomes, to produce a combination known as Benjakul. This Thai traditional medicine is used to treat lung cancer [4], to control elemental imbalances in the body, as a carminative, to balance health and to alleviate flatulence [5]. Plumbagin (5-hydroxy-2-methyl-1,4-naphthoquinone) is a significant bioactive compound found primarily in the roots of *P. indica* [6]. The roots require an extended period to reach maturity to become suitable for medicinal use, typically ranging from one to two years [7]. Furthermore, the fact that whole mother plants are usually destroyed during harvesting, combined with the low rate of vegetative propagation, has led to an inability to meet the demand. Consequently, various methods of plant tissue culture, such as callus culture [8,9], somatic embryogenesis [10], root culture [7] and hairy root culture [11], have been investigated for the purpose of propagating and producing plumbagin. The enhancement of plumbagin production has also been documented through elicitation under aseptic conditions [2].

Elicitation represents a promising and effective biotechnological approach for increasing the concentrations of bioactive compounds in various medicinal plants [12,13]. The production of secondary metabolites through tissue culture has several benefits, including ensuring both quality and yield, allowing for year-round production unaffected by environmental fluctuations, and reducing production time [14]. However, several factors, such as culture type, elicitation exposure period, and elicitor type and dose, can influence the elicitation response. These factors are necessary to identify the optimum conditions for each plant species [13]. Various culture types, including cell and cell suspension cultures [15,16], embryogenic cultures [17], root cultures [18,19] and hairy root cultures [20,21], have been employed to enhance plumbagin yield in *P. indica*. However, no research on plumbagin enhancement has been conducted in shoot cultures as a culture type. According to Jose et al. [2], plumbagin can be harvested from the aboveground portions of this medicinal plant. This indicates that in vitro shoots could be another potential source of plumbagin. In addition, a number of elicitors have been used to enhance plumbagin yield in the genus *Plumbago* in either *P. indica* or *P. zeylanica*. Yeast extract and salicylic acid were chosen for this study due to their effectiveness as biotic elicitors in *P. zeylanica* [22,23,24,25]. Yeast extract is a dead microbe cell or fragment, whereas salicylic acid is a plant hormone. They both are signaling molecules that can enhance the amount of secondary metabolites by inducing the expression of plant defense genes [14]. Studies have indicated that yeast extract and salicylic acid have been demonstrated to enhance the accumulation of bioactive substances in various other medicinal plant species such as *Panax ginseng* [26], *Knautia sarajevensis* [27], *Thymus lotocephalus* [28] and *Musa acuminata* [29]. Yeast extract and salicylic acid, however, have not yet been investigated to ascertain whether they can enhance the amount of plumbagin in regenerated *P. indica* shoots grown in vitro. Hence, the present study is the first to investigate the effects of different concentrations of yeast extract and salicylic acid, and the duration of yeast extract elicitation, on plumbagin accumulation in in vitro-regenerated shoots of *P. indica*. The protocol developed in this study may encourage the large-scale production of plumbagin in the pharmaceutical industry.

## 2. Results

### 2.1. Yeast Extract and Salicylic Acid Elicitation

#### 2.1.1. Fresh and Dry Weights of Regenerated Shoots

After four weeks of culture, the control yielded the maximum fresh and dry weights of 43.48 ± 1.74 mg/shoot and 5.25 ± 0.44 mg/shoot, respectively, as shown in Figure 1. The fresh and dry weights of yeast extract- and salicylic acid-elicited shoots were significantly lower than those of the control. The yeast extract-elicited shoots had fresh weights ranging from 34.53 ± 0.36 to 35.03 ± 1.62 mg/shoot and dry weights ranging from 4.05 ± 0.10 to 4.19 ± 0.12 mg/shoot. In addition, the fresh and dry weights of the salicylic acid-elicited shoots were the lowest at 27.83 ± 1.72–31.47 ± 2.16 mg/shoot and 3.19 ± 0.15–3.33 ± 0.14 mg/shoot, respectively.

#### 2.1.2. Plumbagin Content

A statistically significant difference in the plumbagin contents was detected among treatments (Figure 2). The highest plumbagin contents of 3.88 ± 0.38 and 3.81 ± 0.37% *w*/*w* g dry extract were observed in the 50 and 100 mg/L yeast extract-elicited shoots which were 1.34- and 1.31-fold higher than the control, respectively. The plumbagin contents in the shoots induced by 50–150 μM salicylic acid and 150 mg/L yeast extract (ranging from 2.27 ± 0.38% to 2.70 ± 0.26% *w*/*w* g dry extract) did not exhibit a significant difference compared to the control (2.90 ± 0.08% *w*/*w* g dry extract).

#### 2.1.3. Total Phenolic and Flavonoid Contents, and Antioxidant Activity

The in vitro-regenerated shoots treated with 150 μM salicylic acid exhibited the highest total phenolic content, 36.67 ± 1.92 mg GAE/g dry extract, which was 3.82 times greater than that of the control (9.59 ± 1.95 mg GAE/g dry extract). The shoot treated with salicylic acid concentrations of 50 and 100 μM exhibited total phenolic contents of 31.37 ± 1.26 and 18.08 ± 1.59 mg GAE/g of dry extract, respectively. These levels were 3.27 and 1.88 times greater than that observed in the control. Conversely, yeast extract showed no significant impact on total phenolic content, except when applied at a concentration of 150 mg/L, where it resulted in a value of 17.10 ± 1.77 mg GAE/g dry extract. (Figure 3a).

The contents of total flavonoids in the regenerated shoots treated with yeast extract concentrations ranging from 50 to 150 mg/L (148.57 ± 0.85 and 160.60 ± 7.08 mg CE/g of dry extract) were not significantly different from that of the control (150.87 ± 3.46 mg CE/g of dry extract). Furthermore, the total flavonoid contents of shoots treated with salicylic acid (ranging from 101.79 ± 1.665 to 124.36 ± 3.79 mg CE/g of dry extract) were observed to be lower than that of the control, indicating that salicylic acid had a detrimental effect on total flavonoid content (Figure 3b).

Regarding antioxidant activity, there was no significant difference observed in DPPH radical scavenging activity between the control (56.77 ± 1.29%) and the extracts after treatment with 100 and 150 mg/L yeast extract (52.45 ± 0.49% and 51.62 ± 0.46%, respectively) as well as 150 μM salicylic acid (50.49 ± 4.67%) (Figure 3c). However, the regenerated shoots treated with 50 mg/L yeast extract and 50–100 μM salicylic acid had lower DPPH radical scavenging activity than that of the control. These findings revealed that both yeast extract and salicylic acid had no impact on the antioxidant activity of the *P. indica* shoots regenerated in vitro.

### 2.2. Yeast Extract Elicitation Period

#### 2.2.1. Fresh and Dry Weights of Regenerated Shoots

The fresh and dry weights of the control (ranging from 29.63 ± 1.12 to 45.00 ± 1.20 mg and 3.13 ± 0.18 to 5.07 ± 0.26 mg, respectively) and the shoots elicited with yeast extract (ranging from 28.56 ± 0.60 to 45.30 ± 0.49 mg and 3.10 ± 0.16 to 5.01 ± 0.34 mg, respectively) exhibited an increase over the elicitation period of 3–5 weeks. At each elicitation time, there was no significant difference in fresh and dry weights between the control and the shoots elicited by yeast extract (Figure 4).

#### 2.2.2. Plumbagin Content

The results of the plumbagin content analysis indicated that plumbagin contents ranging from 2.56 ± 0.09 to 2.70 ± 0.13% *w*/*w* g dry extract were observed in the control group at 3–5 weeks of culture (Figure 5), which were not significantly different among treatments. A maximum plumbagin content of 3.22 ± 0.12% *w*/*w* g dry extract was recorded at four weeks in the regenerated shoots elicited with 50 mg/L of yeast extract. This was 1.20-fold higher than that of the control. The 5-week yeast extract-elicited shoots contained the lowest content of plumbagin (1.90 ± 0.01% *w*/*w* g dry extract), which was lower than the control.

## 3. Discussion

In this study of elicitor types and concentrations, the regenerated shoots of *P. indica* elicited with yeast extract showed a slight decrease in biomass compared to the control. Even though there was no significant difference in the biomass of shoots elicited by yeast extract compared to the control during the elicitation period, a trend of a slight decrease in biomass accumulation after elicitation was observed compared to the control, similarly to the first experiments. This could imply that the impact of yeast extract on *P. indica* shoot biomass is minimal. Rahimi et al. [26] reported that when plants are treated with elicitors, they typically exhibit responses such as oxidative burst and the production of reactive oxygen species (ROS). Excessive levels of ROS resulted in damage to membranes, tissue death, the activation of defense mechanisms and, ultimately, a rise in secondary metabolite production [30]. The results from this experiment are in agreement with previous findings, indicating that yeast extract has a negative effect on plant biomass. These reports include shoot cultures of *K. sarajevensis* [27], shoot cultures of *T. lotocephalus* [28] and cell suspension cultures of *Ocimum basilicum* [31]. The salicylic acid-treated shoots showed the lowest biomass. This indicated that salicylic acid had a detrimental effect on shoot biomass, possibly due to high concentrations, which were toxic to the plants, resulting in an imbalanced condition [32]. The results obtained from this finding are consistent with other studies in various plant species such as *M. acuminata* cv. ‘Gros Michel’ [29], *Dioscorea membranacea* [33] and *Primula veris* subsp. *veris* [34], where it was reported that the excessive application of exogenous salicylic acid inhibits plant growth and biomass production. However, the elicitor effect on biomass depends on the plant species, elicitor concentration, duration of elicitation and cultivation method.

Yeast extract improved the amount of plumbagin, particularly at concentrations of 50 and 100 mg/L, while salicylic acid had no effect. Therefore, yeast extract was the preferable elicitor for plumbagin production from in vitro-regenerated shoots of *P. indica*. The components of yeast extract are amino acids, minerals, vitamins, mannose oligosaccharides, β-glucans, chitin and glycogen [35]. According to Fesel and Zuccaro [36], β-glucans and chitin are microbe-associated molecular patterns which can be detected by pattern recognition receptors located within plant cells. This results in the upregulation of genes related to stress and an increase in the production of secondary metabolites. Yeast extract has the benefit of being inexpensive and non-toxic to plants when used as an elicitor. It was found to be the optimum elicitor in many plant species, such as *P. zeylanica* [23,24], *T. lotocephalus* [28] and *Catharanthus roseus* [37]. The antioxidant contents (total phenolic and flavonoid contents) and activity of the non-elicited and elicited regenerated shoots were also analyzed in this study to assess the pharmaceutical value of this plant material. The results indicated that the in vitro-regenerated shoots were shown to have medicinal qualities, but yeast extract and salicylic acid had no effect on total flavonoid or antioxidant activity. This was inconsistent with the finding of Roy and Bharadvaja [23], who reported that yeast extract-elicited root cultures of *P. zeylanica* had higher total phenolic, flavonoid and tannin contents as well as DPPH scavenging activity than the control. This may be due to different plant species, culture type and elicitation process. Salicylic acid at concentrations ranging from 50 to 150 μM increased the total phenolic content of regenerated *P. indica* shoots. This elicitor, a plant hormone, can trigger a protective response called systemic acquired resistance (SAR) when pathogens invade, leading to increases in defense responses and secondary metabolite content [38]. However, salicylic acid adversely affected shoot biomass and did not enhance plumbagin production, suggesting that it was not a suitable elicitor for the in vitro *P. indica* shoots. The presence of plumbagin in the yeast extract-elicited shoots still suggests therapeutic potential even though there was no enhancement in antioxidant content and activity. Further research may focus on other aspects of the elicited shoots, such as their cytotoxic effects on anticancer and anti-inflammatory properties, which could contribute to their medicinal value. Based on these findings, yeast extract was chosen to study the optimal elicitation period in the next experiment, with a focus only on determining plumbagin content.

In addition to elicitation type and concentration, the elicitation period plays a significant role in increasing bioactive compound yield. *P. indica* shoots cultured in vitro were routinely subcultured every 3–5 weeks to promote shoot proliferation. Consequently, an elicitation period ranging from 3 to 5 weeks was implemented for the convenience of harvesting shoots, and the duration of this period was sufficient for the elicitation process. Additionally, the experiment of elicitation type and concentration was performed over a 4-week elicitation period; nevertheless, adjusting the duration, by making it either shorter or longer, could potentially affect the enhancement of plumbagin. These results indicated that the optimum elicitation period for plumbagin in this study was four weeks, whilst the shorter elicitation period of three weeks did not enhance the concentration of this bioactive compound. This may be because there was insufficient time for the expression of related genes. On the other hand, the extended elicitation period of five weeks led to decreased plumbagin content, possibly attributable to degradation or the leaching of bioactive compounds into the culture medium [39]. This finding is consistent with studies on the effect of elicitation period on other plant species such as *D. membranacea* [33], *P. veris* subsp. *veris* [34] and *Rehmannia Glutinosa* [39]. Therefore, our results indicated that the elicitation period had an impact on the plumbagin production of in vitro-regenerated *P. indica* shoots. Although the plumbagin production achieved through yeast extract elicitation from this study showed potential, further investigation into various factors, including temperature, nutrient composition, medium pH and status, or alternative approaches such as precursor feeding, is necessary to improve yields. 

It has been previously documented that plumbagin production can be enhanced, particularly in the genus *Plumbago*. Numerous *Plumbago* culture types, including root, hairy root, callus and cell suspension cultures, have been examined [15,16,17,18,19,20,21,40], but the in vitro-regenerated shoots of *P. indica* have not been the subject of this kind of investigation. One advantage of application with the regenerated shoots is maintaining genetic stability. In addition, the elicitation procedure was convenient as it could be completed in a single step by culturing nodal explants in a culture medium containing an elicitor. Recent research has employed regenerated *P. zeylanica* shoots to enhance plumbagin production through elicitation [25]. Numerous plant species have bioactive compounds that accumulate in their field-grown roots or underground stems. The utilization of shoot cultures has resulted in an increase in bioactive compounds in such plants as *D. membranacea* [33], *Alpinia zerumbet* [41], *Bosenbergia rotunda* [42] and *D. zingiberensis* [43]. The findings from this study indicate that the in vitro-regenerated shoots of *P. indica* elicited with yeast extract offer another potential source of plumbagin and promising opportunities for pharmaceutical applications by reducing the processing duration, and ensuring the quality and sustainable sourcing of plumbagin production. This may result in the development of novel medications or supplements aimed at relieving symptoms of diseases. However, using in vitro-regenerated shoots as plant material for the elicitation process may encounter limitations in scalability and physiological abnormality. Further investigation is required to scale up shoot materials and increase plumbagin yield for large-scale applications, such as in a bioreactor system. Studying the optimization of bioreactor culture conditions, including the composition of the culture medium, the temperature and the intensity and quality of light, is necessary to scale up plant materials, maintain physiological functions, ensure genetic stability and improve plumbagin production.

## 4. Materials and Methods

### 4.1. Plant Material and Culture Conditions

Young lateral shoots of *P. indica* were collected from the botanic garden of the Department of Thai Traditional and Alternative Medicine, Ministry of Public Health, in Nonthaburi province, Thailand. Single-node explants were trimmed to lengths of approximately 1–1.5 cm, washed with tap water and subsequently subjected to a two-step surface sterilization process. Initially, the procedure involved immersing the explants in a 10% (*v*/*v*) sodium hypochlorite solution with a few drops of Tween 20 for 15 min, followed by a subsequent immersion in a 5% (*v*/*v*) solution for 10 min. After being rinsed twice with sterilized distilled water, the nodal explants were then transferred to MS medium supplemented with 1 mg/L benzyladenine (BA) to induce shoot formation. The shoots regenerated in vitro were subcultured to fresh medium every four weeks. The cultures were kept at a temperature of 25 ± 2 °C under a photoperiod of 16 h light and 8 h darkness for maintenance. Culture medium used for plumbagin elicitation was contained 1 mg/L BA, 30 g/L sucrose and 8 g/L agar. The pH of the medium was adjusted to 5.6–5.8 using 1 N NaOH prior to autoclaving at 121 °C for 15 min.

### 4.2. Elicitor Preparation

Yeast extract and salicylic acid were used to determine the elicitation effect in the in vitro-regenerated *P. indica* shoots. The culture medium was supplemented with 50, 100 and 150 mg/L yeast extract (Merck, Darmstadt, Germany) after it had been dissolved in distilled water. Salicylic acid (BDH Prolabo, Leuven, Belgium) was prepared as described by Singh et al. [24]. Salicylic acid powder was dissolved in the minimal amount of 1 N NaOH and then diluted with distilled water to produce a stock solution at a concentration of 0.1 M. A 0.22 µm syringe filter was used to sterilize the salicylic acid solution before it was added to the culture medium at final concentrations of 50, 100 and 150 µM.

### 4.3. Elicitation with Yeast Extract and Salicylic Acid

The shoots regenerated in vitro were used and cut into nodal segments, each approximately 1–1.5 cm in length. The nodal explants were cultured on MS medium supplemented with 1 mg/L BA in combination with 50, 100 and 150 mg/L yeast extract or 50, 100 and 150 µM salicylic acid for a duration of four weeks. A control was established using MS medium supplemented with 1 mg/L BA. The study followed a completely randomized design (CRD) comprising seven treatments and three replications. After four weeks of elicitation, the regenerated shoots of each treatment were collected, and the fresh weight of the shoots was measured. The dry weight of the shoots was measured after 48 h of drying at 50 °C. Examination was carried out on both control and elicited shoots to assess their plumbagin, total phenolic and flavonoid levels, along with their antioxidant-scavenging activity.

### 4.4. Yeast Extract Elicitation Period

The in vitro nodal segments were cultured on MS medium supplemented with 1 mg/L BA in combination with either 0 or 50 mg/L yeast extract for durations of three, four and five weeks. The study was conducted using a CRD consisting of six treatments and three replications. Following harvesting of the regenerated shoots from each treatment, the fresh weight of the shoots was recorded. The shoots were subsequently dried at 50 °C for 48 h, and their dry weight was recorded before analyzing the plumbagin content of each treatment.

### 4.5. Bioactive Compound Analysis

#### 4.5.1. Plant Extract Preparation

The dried samples from each treatment were ground and the powder was subjected to extraction using 95% ethanol at a sample:ethanol ratio of 1:3 for a three-day period. The process of maceration was performed thrice. The extracts from the shoots were combined, filtered and evaporated to complete dryness using a rotary evaporator under reduced pressure [44]. The dried extracts were preserved at −20 °C for subsequent analysis.

#### 4.5.2. Preparation of Standard for Determination of Plumbagin Content

A precise amount of 1 mg standard plumbagin was weighed and subsequently dissolved in 1 mL of acetonitrile (HPLC grade) to create a stock solution with a concentration of 1 mg/mL. The stock solution was diluted in parallel to produce standard solutions with concentrations of 10, 25, 50, 100, 150, 200, 400 and 800 μg/mL, which were used for further analysis.

#### 4.5.3. Determination of Plumbagin Content by Reversed Phase–High Performance Liquid Chromatography (RP–HPLC)

The method for analyzing plumbagin content was adjusted based on the procedure outlined by Itharat and Sakpakdeejaroen [44] and Kuropakornpong et al. [45]. A total of 10 mg of crude extract was dissolved in 1 mL of acetonitrile (HPLC grade). The combination underwent sonication for a duration of 15 min, after which it was filtered through a 0.45 μM membrane filter before being injected into the RP-HPLC system. This system consisted of a reversed-phase Luna C18(2), a 100A column (250 × 4.60 mm, particle size 5.0 micron) with a guard column (Phenomenex, Inc., Torrance, CA, USA) and an Agilent 1200 Infinity system (Agilent Technologies, Santa Clara, CA, USA) that included a solvent degasser (G1322A), a quaternary solvent pump (G1311A), an autosampler (G1329A), a column oven (G1316A) and a photodiode array detector (G1315D). Ten-microliter samples of the solution were injected and subjected to gradient elution with a mixture of acetonitrile and water. The gradient elution conditions were as follows: 0 min, 40:60; 30 min, 50:50; 40 min, 95:5; 45 min, 40:60; and 50 min, 40:60 with a flow rate of 1.0 mL/min. Plumbagin concentration was determined at 256 nm (Figure 6). The linear concentration range was 10–800 μg/mL. The chromatogram analysis was performed using ChemStation software Rev. B.04.01 (Agilent Technologies, Santa Clara, CA, USA).

#### 4.5.4. Determination of Total Phenolic and Flavonoid Contents, and Antioxidant Activity

The Folin–Ciocalteu method was employed to determine the total phenolic content. The analytical procedures were carried out according to the methods previously outlined by Jirakiattikul et al. [29]. Briefly, a combination of 1 mg of plant extract with 1 mL of absolute ethanol was subjected to one minute of sonication. An amount of 20 µL of sample was combined with 100 µL of diluted Folin–Ciocalteu reagent and 80 µL of a 7.5% (*w*/*v*) sodium carbonate solution. The solution was allowed to incubate at a temperature of 25 °C for a duration of 30 min. The absorbance value at 765 nm was read using a spectrophotometer (Thermo Fisher Scientific: Multiskan GO 1510-04583C, Waltham, MA, USA). The content of total phenolic compounds was presented as milligrams of gallic acid equivalent per gram of dry extract (mg GAE/g dry extract).

The aluminum chloride colorimetric method was used to assess the flavonoid content, as described by Jirakiattikul et al. [29]. A solution of plant extract at a concentration of 1 mg/mL was prepared, and 500 µL of this solution was combined with 75 µL of 5% (*w*/*v*) NaNO_2_ for 6 min. Subsequently, 150 µL of AlCl_3_ solution at a concentration of 10% (*w*/*v*) was added. The reaction continued for 5 min, following which 500 µL of 1 M NaOH and 275 µL of distilled water were included. The solution was then left to stand for a period of 15 min. The absorbance value at 510 nm was read using a spectrophotometer. The content of total flavonoids was presented as milligrams of catechin equivalent per gram of dry extract (mg CE/g dry extract). 

A DPPH (2,2-diphenyl-1-picryl-hydrazyl-hydrate) radical scavenging assay was utilized to evaluate the antioxidant activity of the in vitro-regenerated shoots obtained from *P. indica*. The procedure was adapted from Jirakiattikul et al. [29]. The plant extracts, initially dissolved in 1 mL of absolute ethanol at a concentration of 1 mg/mL, were subsequently diluted to concentrations of 200 µg/mL. In a 96-well microplate, 100 µL of 200 µg/mL of extract along with 100 µL of 6 × 10^−5^ M DPPH in absolute ethanol were combined and allowed to incubate in darkness at room temperature for 30 min. Butylated hydroxyltuluene (BHT) and absolute ethanol were used as the positive and negative controls, respectively. The absorbance was assessed at 520 nm utilizing a spectrophotometer. The percentage inhibition was determined using the formula % inhibition = [(Abs _control_ − Abs _sample_)/Abs _control_] × 100, where Abs _control_ represents the absorbance of the control, and Abs _sample_ represents the absorbance of the test sample.

### 4.6. Statistical Analysis

Data were analyzed by One-way ANOVA and the means were compared utilizing Tukey’s Honestly Significant Difference (HSD) test at a significance level of 0.05 using SPSS software version 23.

## 5. Conclusions

We found that the production of plumbagin (3.22 ± 0.12% *w*/*w* g dry extract) was promoted by yeast extract at a concentration of 50 mg/L for a duration of four weeks. This finding suggested that the elicited shoots of *P. indica* grown in vitro offer shorter processing times and reliable plumbagin production for pharmaceutical uses. Further investigation is required to improve yields by studying factors such as temperature, nutrient composition, medium pH and status. The bioreactor culture conditions require optimization for scaling up plant materials and increasing plumbagin yield. Additionally, assessing its cytotoxic effects is essential for understanding its medicinal potential.

## Figures and Tables

**Figure 1 plants-13-01450-f001:**
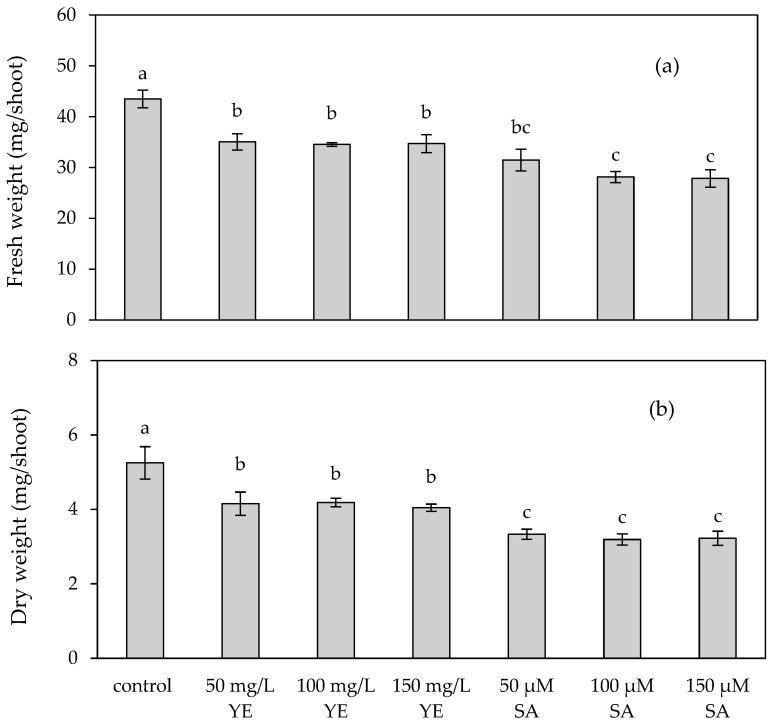
(**a**) Fresh weight and (**b**) dry weight of in vitro *Plumbago indica* shoots after regeneration on MS medium supplemented with 1 mg/L BA (control) or in combination with different concentrations of yeast extract (YE) and salicylic acid (SA) for 4 weeks. Different letters above the columns indicate a significant difference (*p* < 0.05), and error bars indicate ± SD.

**Figure 2 plants-13-01450-f002:**
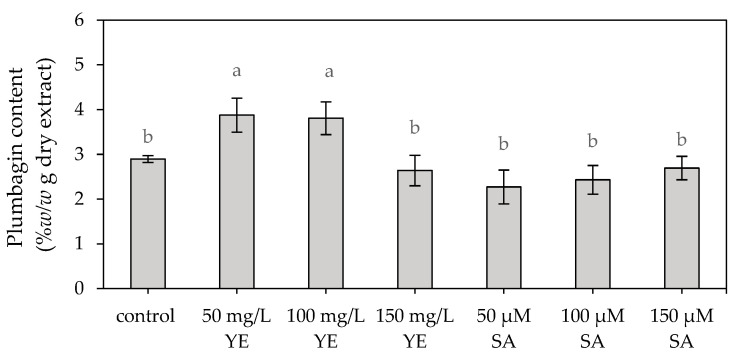
Plumbagin content of in vitro *Plumbago indica* shoots after regeneration on MS medium supplemented with 1 mg/L BA (control) or in combination with various concentrations of yeast extract (YE) and salicylic acid (SA) for a duration of 4 weeks. Different letters above the columns indicate a significant difference (*p* < 0.05), and error bars represent ± SD.

**Figure 3 plants-13-01450-f003:**
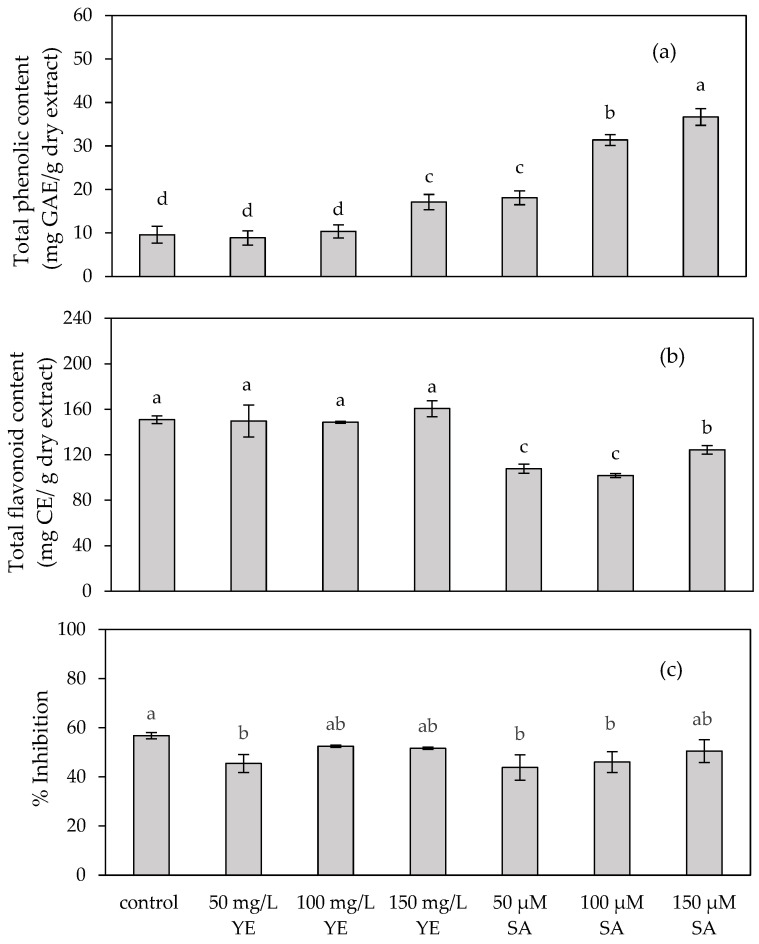
(**a**) Total phenolic content, (**b**) total flavonoid content and (**c**) % DPPH inhibition of in vitro *Plumbago indica* shoots after regeneration on MS medium supplemented with different concentrations of yeast extract (YE) and salicylic acid (SA), plus control for 4 weeks, where GAE is gallic acid equivalents, CE is catechin equivalents, different letters above the columns indicate a significant difference (*p* < 0.05) and error bars indicate ± SD.

**Figure 4 plants-13-01450-f004:**
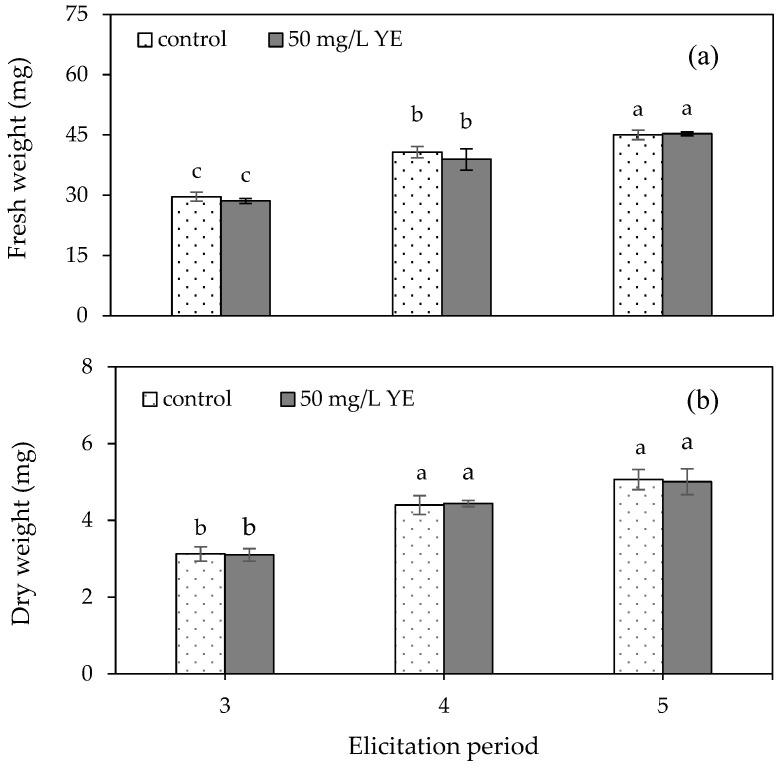
(**a**) Fresh weight and (**b**) dry weight of in vitro *Plumbago indica* shoots after regeneration on MS medium supplemented with 0 (control) and 50 mg/L yeast extract for three, four and five weeks. Different letters above the columns indicate a significant difference (*p* < 0.05), and error bars indicate ± SD.

**Figure 5 plants-13-01450-f005:**
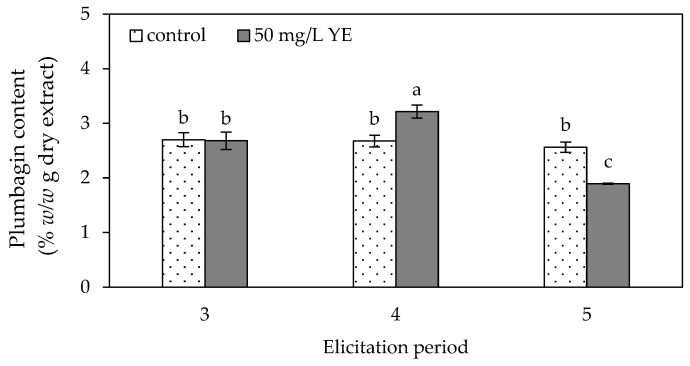
Plumbagin content of *Plumbago indica* shoots regenerated in vitro on MS medium supplemented with 0 (control) and 50 mg/L yeast extract (YE) for three, four and five weeks. Different letters above the columns indicate a significant difference (*p* < 0.05), and error bars indicate ± SD.

**Figure 6 plants-13-01450-f006:**
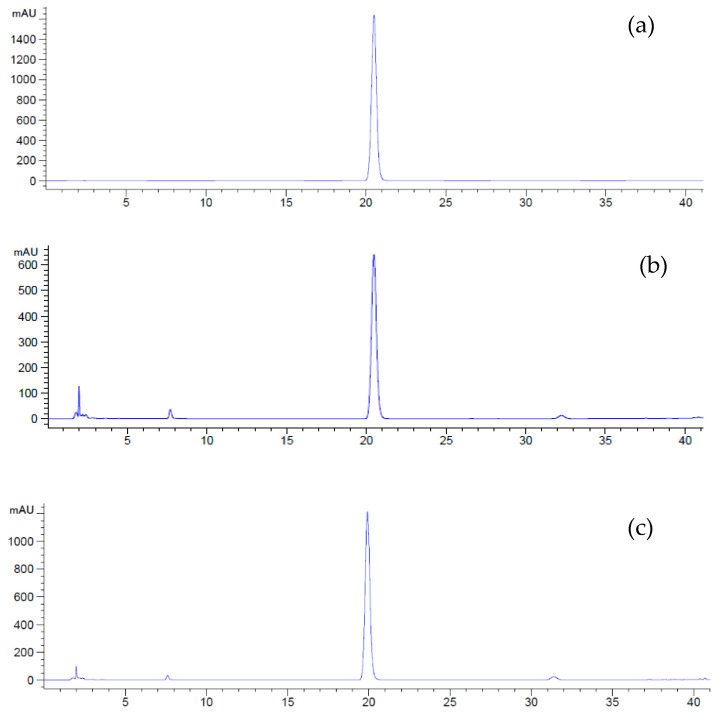
HPLC chromatograms of plumbagin content at 256 nm: (**a**) the standard solution (200 μg/mL), (**b**) the in vitro-regenerated shoots cultured on MS medium supplemented with 1 mg/L BA or control and (**c**) the in vitro-regenerated shoots treated with 50 mg/L yeast extract for 4 weeks.

## Data Availability

The data are contained within the article.

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
