# Peer review of "Enhancement of Plumbagin Production through Elicitation in In Vitro-Regenerated Shoots of Plumbago indica L."

_plants, 2024, doi:10.3390/plants13111450_

Round 1
Reviewer 1 Report
Comments and Suggestions for Authors
This is a well-presented, straightforward study on the effects of two elicitors, yeast extract and salicylic acid, on the in vitro shoot culture of a medicinal plant Plumbago indica. The authors studied the content of the main bioactive compound, plumbagin, along with total phenolics, flavonoids, and antiradical activity. The aim of the study fits the journal perfectly, the data are new and would be of interest for a broad audience of plant biotechnologists and specialists in phytochemical production. All methods are relevant and performed with necessary controls and sufficient number of replications. The paper is overall of very good quality and I recommend accepting it for publication after revision. There is one mismatch between the data presented in the figures which I highlights below and a few minor comments that I ask the authors to address.
Major comment:
The results showed in Fig. 4 are controversial to those shown in Figs 1. According to Fig. 4, both dry and fresh weights of the shoots under elicitation with 50 mg/L YE are not different from those in control group without elicitation after 4 weeks of treatment. This contradicts to data given in Fig. 1 where the same concentration of YE (50 mg/L) caused significant decrease in both fresh and dry weights after the same treatment period (4 weeks). Please check which figure is correct. Apologies if I misunderstood your treatments; in this case, please explain the difference between these datasets more clearly.
Minor comments:
Results. 2.1.2. Plumbagin content. – Additionally to all information that you give about plumbagin content, please add the maximum amount of plumbagin achieved in this study calculated based on g dry weight of plant material rather than g dry weight of the extract. This will allow the readers to compare your data with outcomes of other similar studies. Please do the same for the abstract.
Minor comments:
Lines 164-169. “Regarding antioxidant activity, there was no significant difference observed in antioxidant activity between the control (56.77 ± 1.29%) and extracts after treatment with yeast extract (ranging from 45.42 ± 3.66% to 52.45 ± 0.49%) and salicylic acid (ranging from 43.80 ± 5.23% to 50.49 ± 4.67 %) (Figure 3c). The findings revealed that both yeast extract and salicylic acid had no impact on the antioxidant activity of the shoots regenerated in vitro of P. indica.” – I can see from Fig. 3c that there is a significant difference in antiradical activity between shoots treated with 20 mg/L YE and 50-100 µM SA. Please check if the description of these data in the text is correct.
Lines 254-256. “Numerous Plumbago culture types, including root, hairy root, callus and cell suspension cultures have been examined, but the in vitro regenerated shoots of P. indica have not been the subject of the same investigation.” – Please add a reference (references) after this statement. You may also refer to one of the recent reviews: https://doi.org/10.1007/s00253-023-12511-6
Line 289. “However, there was no significant difference in the biomass of shoots elicited by yeast extract compared to the control during the elicitation period.” – This contradicts to the results shown in Fig. 1.
Line 299. Better to replace “optimum” with “preferable”. To make conclusion about optimum elicitor you have to test multiple elicitors, not just two of them.
Line 330. “cultured conditions” – change to “culture conditions”
Line 331. “Young lateral shoots of P. indica were collected from the Ministry of Public Health in Nonthaburi, Thailand.” – Please mention the institute name (if relevant) and if the plants were collected from greenhouse or botanic garden, or a lab.
Line 343. “for15 min” – space is missing “for 15 min”
Line 346. “shoot” -> “shoots”
Lines 346 and 352 – Please mention if these are final concentrations in the medium
Lines 424-434 – DPPH analysis - Please indicate what was used as a control sample
Conclusion – Please add the maximum concentration of plumbagin that was achieved in the study
Reviewer 2 Report
Comments and Suggestions for Authors
The study investigated the effects of different elicitors and elicitation periods on plumbagin content in in vitro regenerated shoots of Plumbago indica. Major adjustment is required. In my opinion author has fails to give rationale of the their findings.
There have been numerous studies published on this topic. On same or different species and with same or different elicitation methods. This raises questions about the novelty of their research.
In this scenario; author need to explain why and how their work is unique and different from previously published studies on the same subject.
For example; Gangopadhyay, M., Dewanjee, S., & Bhattacharya, S. (2011). Enhanced plumbagin production in elicited Plumbago indica hairy root cultures. Journal of Bioscience and Bioengineering, 111(6), 706-710. https://doi.org/10.1016/j.jbiosc.2011.02.003
Katoch, K., Gupta, S., Gupta, A.P. et al. Biotic elicitation for enhanced production of plumbagin in regenerated shoot cultures of Plumbago zeylanica using response surface methodology. Plant Cell Tiss Organ Cult 151, 605–615 (2022). https://doi.org/10.1007/s11240-022-02375-5.
Komaraiah, P., Naga Amrutha, R., Kavi Kishor, P., & Ramakrishna, S. (2002). Elicitor enhanced production of plumbagin in suspension cultures of Plumbago rosea L. Enzyme and Microbial Technology, 31(5), 634-639. https://doi.org/10.1016/S0141-0229(02)00159-X
Line 14; mention the type of elicitors.
Line 19, 23; add the control value.
Add significance and any potential limitations of the study at the end of the abstract. These details would enhance the clarity and reliability of the findings presented in the abstract.
Line 29; what is the geographical status? IUCN category? According to The Plant List? Provide reference and web link.
Line 33; What is a GRAS (Generally recognized as safe) status of Plumbago indica? What is the recommended concentration of P. indica for human consumption? The Authors need to mention this concentration as per safety regulatory agencies like US-FDA, FAO/WHO and Council of Europe and the United Kingdom, Food Additive and Contaminants Committee etc.
Line 149; The total phenolic content in shoots treated with salicylic acid was significantly higher compared to the control. What is the possible reason? considering that salicylic acid elicitation resulted in lower fresh and dry weights.
Line 2016; The study mentions that the highest plumbagin contents were observed in shoots elicited with yeast extract. The study reports that yeast extract elicited shoots had lower fresh and dry weights compared to the control. How the both findings are correlated? Discuss the possible reason.
Line 252; Discussion; The study mentions that in vitro regenerated shoots of P. indica were used for plumbagin production, citing advantages such as genetic stability and convenience in elicitation. However, author fail to provide potential limitations or challenges associated with using in vitro regenerated shoots compared to other culture types (e.g., root cultures, callus cultures) in terms of scalability, maintenance of physiological functions, and potential genetic variations? These scenarios are needed to be discuss properly.
The study emphasizes the pharmaceutical potential of in vitro regenerated shoots, yet the lack of significant changes in antioxidant activity following elicitation raises questions about the medicinal value of the elicited shoots. How author is correlating this finding for the potential therapeutic application of P. indica?
Line 287-293; author is unable to explain the rationale for the observation for example; yeast extract elicited shoots showed a slight decrease in biomass when compared to the control while salicylic acid elicited shoots showed the lowest biomass.
Line 318; The optimum elicitation period for plumbagin production was determined to be four weeks. Could you discuss the rationale behind choosing this specific period and whether longer-term studies (beyond five weeks) were considered to investigate potential trends in plumbagin accumulation or stability over time?
In addition; discussion is lacking in the translational potential of the findings from this study and areas for future research to bridge the gap between in vitro studies and real-world applications in pharmaceutical or agricultural industries?
Highlight any other factors that might influence plumbagin production, such as the age or type of culture used, environmental conditions, or the specific genetic makeup of the plant material.
Conclusion need to be re-written by considering the followings; Expand on the practical applications of these findings; Highlight any other factors that might influence plumbagin production; Suggest avenues for future research based on the current findings.
Comments on the Quality of English Language
The manuscript needs to be revised in terms of English and grammar.
Round 2
Reviewer 1 Report
Comments and Suggestions for Authors
I have reviewed the revised version, and found that the authors corrected nearly all controversial statements in the text. I suggest one minor correction in the following sentence:
Initial Comments 5: Line 289. “However, there was no significant difference in the biomass of shoots elicited by yeast extract compared to the control during the elicitation period.” – This contradicts to the results shown in Fig. 1.
Response 5: This sentence is correct. The response to the comment is the same as response 1. However, we have included an additional sentence to enhance clarity on page 6, lines 276-278.
"Even though there was no significant difference in the biomass of shoots elicited by yeast extract compared to the control during the elicitation period, a similar trend of slight decrease was observed compared to the control"
I suggest the following change to this sentence:
"Even though there was no significant difference in the biomass of shoots elicited by yeast extract compared to the control during the elicitation period, a trend of slight decrease in biomass accumulation after elicitation was observed compared to the control, similarly to the first experiments"
With this correction, paper can be accepted for publication.
Reviewer 2 Report
Comments and Suggestions for Authors
The author has incorporated all the suggestions. The manuscript can be considered now for publication.
